# Merging Satellite and Gauge Rainfalls for Flood Forecasting of two Catchments under Different Climate Conditions

**Xinyi Min** [1,2]**, Chuanguo Yang** [1,2,*] **and Ningpeng Dong** [1,2,3]

1    College of Hydrology and Water Resources, Hohai University, Nanjing 210098, China;
     fowlervivian66@gmail.com (X.M.); dnp908393893@gmail.com (N.D.)
2    State Key Lab of Hydrology-Water Resources and Hydraulic Engineering, Hohai University,
     Nanjing 210098, China
3    Department of Water Resources, China Institute of Water Resources and Hydropower Research,
     Beijing 100038, China
*    Correspondence: cgyang@hhu.edu.cn

**Abstract:** As satellite rainfall data has the advantages of wide spatial coverage and high spatial and temporal resolution, it is an important means to solve the problem of flood forecasting in ungauged basins (PUB). In this paper, two catchments under different conditions, Xin'an River Basin and Wuding River Basin, were selected as the representatives of humid and arid regions, respectively, and four kinds of satellite rainfall data of TRMM 3B42RT, TRMM 3B42V7, GPM IMERG Early, and GPM IMERG Late were selected to evaluate the monitoring accuracy of rainfall processes in the two catchments on hourly scale. Then, these satellite rainfall data were respectively integrated with the gauged data. HEC-HMS (The Hydrologic Engineering Center's-Hydrologic Modeling System) model was calibrated and validated to simulate flood events in the two catchments. Then, improvement effect of the rainfall merging on flood forecasting was evaluated. According to the research results, in most cases, the Nash–Sutcliffe efficiency coefficients of the simulated streamflow from initial TRMM (Tropical Rainfall Measuring Mission) and GPM (Global Precipitation Measurement) satellite rainfall data were negative at the two catchments. By merging gauge and TRMM rainfall, the Nash–Sutcliffe efficiency coefficient is mostly around 0.7, and the correlation coefficient is as high as 0.9 for streamflow simulation in the Xin'an River basin. For the streamflow simulated by merging gauge and GPM rainfall in Wuding River basin, the Nash–Sutcliffe efficiency coefficient is about 0.8, and the correlation coefficient is more than 0.9, which indicate good flood forecasting accuracy. Generally, higher performance statistics were obtained in the Xin'an River Basin than the Wuding River Basin. Compared with the streamflow simulated by the initial satellite rainfalls, significant improvement was obtained by the merged rainfall data, which indicates a good prospect for application of satellite rainfall in hydrological forecasting. In the future, it is necessary to further improve the monitoring accuracy of satellite rainfall products and to develop the method of merging multi-source rainfall data, so as to better applications in PUB and other hydrological researches.

**Keywords:** satellite rainfall; GPM IMERG; TRMM 3B42; merging data; flood forecasting

---

## 1. Introduction

Flood disaster is one of the most serious natural disasters in China which not only poses a threat to the safety of people's life and property but also disturbs the construction of a harmonious society [1].The effects of flood disaster on human society economic development include human health, life property, economic income, living environment, mental disorder, ecological environment, and other

direct and indirect impacts [2].The research of flood forecasting, so as to avoid huge losses by flood disasters has become an important part of hydrological study. Recently, flood prevention engineering has been strengthened in medium and small basins, and the construction of nonengineering facilities such as hydrologic monitoring and flood forecasting were also improved to provide decision basis for flood control command [3]. The floods in medium and small basins have the characteristics of high precipitation rate, short time period, and rapid routing speed [4], which are remarkably different from the flood process of large rivers, so the simulation method cannot follow the calculation method of design flood for large rivers [5]. At present, the rainfall-runoff models are widely used to flood forecasting in medium and small basins, and the simulation results are acceptable.

Rainfall is one of the important parts of the global water cycle [6], and it is related to the atmospheric circulation and climatic variations [7]. Also, it is vital to the weather forecast [8], establishment of hydrological model [9], and disaster monitoring [10]. At present, the rainfall data in hydrological forecasting research mainly come from the rainfall stations [11], but the distribution of rainfall stations is often uneven because of economic and terrain constraints. In addition, rainfall stations are also hard to meet the theoretical distribution density requirements [12]. Now, there are still many basins without sufficient rainfall data in the world. That being said, hydrologists and managers have faced a notable challenge on flood forecasting in ungauged basin [13].

Recently, with the improvement of the remote sensing technology and inversion algorithm based on satellite data [14], rainfall monitoring by satellites has got more and more attention. Satellite rainfall have the advantages of large continuous coverage, high spatial, and temporal resolutions [15], so it gradually occupies a place in the field of hydrological research. Now, the TRMM (Tropical Rainfall Measuring Mission) and GPM (Global Precipitation Measurement) satellite rainfalls are used widely, which were jointly developed by NASA (US National Aeronautics and Space Administration) and Japan Aerospace Exploration Agency (JAXA) and the data can be downloaded from NASA's Precipitation Measurement Missions website (https://pmm.nasa.gov/); many researchers have carried out a series of results on them in recent years. Liu [16] found that TRMM and GPM satellite rainfall have high accuracy, and the spatial resolution and precision of the GPM satellite rainfall is higher in southeast China than that in rest of China. Wu [17] used 224 rainfall stations in the Yangtze river basin to evaluate the accuracy of GPM IMERG v5 and TRMM 3B42 v7. The result shows that the precision of GPM IMERG (Integrated Multi-satellitE Retrievals for GPM) is superior to TRMM 3B42 in rainfall monitoring, but its ability in retrieving heavy rainfall and in detecting complex terrain environment is poor. In terms of the topographic influence on the GPM and TRMM, Xu [18] evaluated their accuracy in the southern Tibetan plateau. They found that there was a negative correlation between accuracy and altitude, and GPM satellite has more advantages in the plateau area compared to the TRMM data, such as lower false alarming and better detecting ability for light rainfall (0–5 mm/d). Zhang [19] assessed the precision of GPM IMERG, TRMM 3B42, and CMORPH (Climate Prediction Center morphing method) satellite rainfall products in the Tianshan region, where both the precipitation and the rainfall stations are unevenly distributed. The results show that GPM has better performance than TRMM and CMORPH in estimating daily precipitation, and its monitoring accuracy in middle and high latitude dryland is higher than the latter, but it is not good enough in tropical mountainous areas. In addition, Zhang [20] found that the accuracy of GPM IMERG is better than that of TRMM TMPA at multiple time scales, but there is still room for improvement.

With the improvement of the satellite technology for monitoring rainfall, it is becoming possible to use satellite rainfall in flood forecasting. Wu [21] chose GPM IMERG rainfall as the input data of the THREW (Representative Elementary Watershed) hydrological model. The results indicated that it is of great significance for real-time flood forecasting in areas without data if the GPM satellite rainfall was calibrated by a few rainfall stations in the region. Li [22] selected the Three Gorges Reservoir as a typical domain and proved the application value of high-resolution radar observation information in flood simulation by a distributed hydrological simulation system. Gilewski [23] evaluated GPM IMERG satellite rainfall by the HEC-HMS (The Hydrologic Engineering Center's-Hydrologic Modeling

System) model for mountain rivers, the model was developed by the Water Resources Research Center of the United States Army Engineering Corps in the 1990s (https://www.hec.usace.army.mil/software/hec-hms/). The results showed that the gauge-adjusted data was a reliable data source, and it was useful in flood forecasting. Tekeli [24] conducted flood forecasting with TRMM 3B42RT in Saudi Arabia and developed a flood forecast system based on 3B42RT rainfall, which indicated that the satellite rainfall was suitable for flood forecasting. Based on the current research results, it can be found that satellite rainfall is of potential practicability in areas without data and has prospects for application in runoff simulation. However, satellite rainfall data needs to be compared and corrected with ground stations at different areas, and there is still further development for their use in flood forecasting.

The TRMM and GPM satellite rainfall have good accuracy at large time and spatial scales, but they are greatly impacted by local climate and terrain conditions [25]. The present studies mainly focused on large basins and daily–monthly time scales, for example, the southeast China and Yangtze River basins mentioned above, which cover a large area and have numerous tributaries and precipitation sites. Hydrological application of such satellite rainfalls needs to pay more attention to flood simulation in medium and small basins, such as Wuding River Basin, which covers only a few thousand sq km and contains fewer precipitation sites. This paper evaluates accuracy of the satellite rainfall TRMM 3B42 and GPM IMERG in both the Xin'an River catchment and the Wuding River catchment, which reflect humid and semiarid region climate conditions, respectively. Then, the gauged rainfall data are merged with the satellite rainfall by a proposed method. The HEC-HMS model is used to simulate streamflow by gauged rainfall and satellite rainfall before and after correction, which verify the feasibility of satellite rainfall in flood forecasting at medium and small catchments.

## 2. Study Area and Data

### 2.1. Study Areas

#### 2.1.1. Xin'an River Catchment

Xin'an River catchment is located in the east Asia humid monsoon area between 117°37′~118°56′ E, 29°25′~30°16′ N. The river originates from Wugujian Mountain at the junction of Xiuning County, Anhui province, and Jiangi province and routes into Qiandao Lake in Zhejiang province [26]. It borders Zhejiang province in the east and southeast and Jiangxi province in the west and southwest. The length of the main stream of the Xin'an River is about 359 km, and the study area is about 6219 km$^2$ in this paper (Figure 1). The multi-year average precipitation in the basin is 1700.4 mm, which has a remarkably uneven seasonal distribution. The rainfall in the flood period (April to September) accounts for 70% of the total annual rainfall [27], which are prone to causing mountain flood disasters.

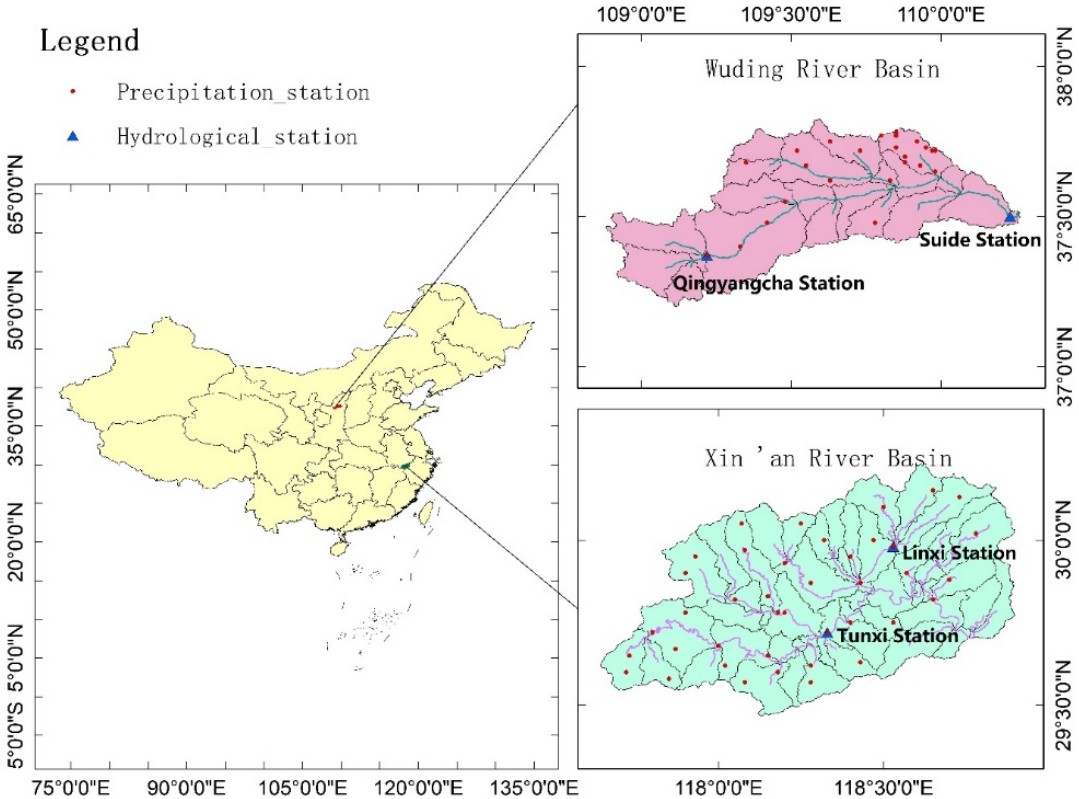

**Figure 1.** Precipitation station and hydrological station in the Xin'an River catchment and the Wuding River catchment.

The sub-watershed division and river networks were carried out by the preprocessing module of the HEC-HMS model in the Xin'an River catchment, as shown in Figure 1. The observed streamflow data of Tunxi (2689 km$^2$) and Linxi (589 km$^2$) stations for flood events in the period 2007–2015 except the year 2014 was collected from Statistical Bulletin of Chinese Hydrology.

2.1.2. Wuding River Catchment

Wuding River catchment is located in the middle region of the Yellow River basin [28], between 108°18'~111°45' E, 37°14'~39°35' N, where it spreads a vast expanse of a loess plateau. The Wuding River belongs to the first-order branch of the Yellow River, which is the largest river in Yulin city of Shaanxi province. It originated from the northern area of Baiyu Mountain at the junction of Jingbian, Dingbian, and Wuqi counties in Shaanxi province and then flows into the main channel of the Yellow River through Hengshan, Yulin, Mizhi, and Suide counties in the province. The river has a total length of 491 km [29], and the study area of this paper is about 4000 km$^2$ (also seen in the Figure 1). The river usually has a small flow, but high intensity rainfalls can also occur in a short period of time in the summer [30], which cause flash floods with high sediment concentration due to severe erosive sediment yield in the catchment.

With the same spatial analysis, we obtained the sub-watershed division and river networks in Wuding River catchment, as shown in the Figure 1. The observed streamflow data of the flood events at Qingyangcha (662 km$^2$) and Suide (3893 km$^2$) stations in the period 2006–2014 were also collected from Statistical Bulletin of Chinese Hydrology.

*2.2. Satellite and Gauged Rainfall Data*

The TRMM (Tropical Rainfall Measuring Mission) and GPM (Global Precipitation Measurement) satellite rainfalls are used in this study. TRMM is the satellite project to undertake global rainfall monitoring [31], and GPM was built upon the TRMM project. They were jointly developed by

NASA (US National Aeronautics and Space Administration) and Japan Aerospace Exploration Agency (JAXA) [32]. TRMM 3B42 V7, TRMM 3B42RT, and GPM IMERG "early run" and IMERG "late run" were downloaded from NASA's Precipitation Measurement Missions website (https://pmm.nasa.gov/). Both TRMM 3B42 products have a 0.25° × 0.25° spatial resolution and 3-hour time step, while the GPM IMERG products have 0.1° × 0.1° spatial resolution and 0.5-hour time step.

The TRMM 3B42RT provides near-real-time precipitation data, and the data supplied by TRMM 3B42V7 is non-real time. The difference between them is that the TRMM 3B42RT near-real-time data was bias corrected using the CCA (Climatologic Calibration Analysis), whereas the TRMM 3B42V7 non-real time data was corrected by merging with gauge data from a few meteorological stations [33].

In addition, the system of GPM is run several times for each observation time, first giving a quick estimate (IMERG "early run") and successively providing better estimates as more data arrive (IMERG "late run"). The final step uses monthly gauge data to create research-level products (IMERG "final run"). Flood forecasting requires real-time precipitation monitoring as much as possible. If we use the final run data, the longer lag time of the data is not beneficial to flood forecasting. IMERG "final run" is also difficult to meet the needs of flood forecasting in medium and small basins due to the quality of gauge data. For example, the gauge density is not enough or the gauge data are on monthly scale not hourly scale. Since the GPM satellites launched in 2014, GPM IMERG data were only adopted after 2014.

The observed hourly rainfall data used in this paper are also extracted from the Statistical Bulletin of Chinese Hydrology. There are respectively 39 rainfall stations and 25 rainfall stations distributed in the Xin'an River catchment and Wuding River catchment as shown in the Figure 1.

## 3. Evaluation Indices and Merging Method for Rainfall Data

### 3.1. Evaluation Indices

Correlation Coefficient (CC), Root Mean Squared Error (RMSE), Mean Error (ME), Mean Absolute Error (MAE), and Relative Bias (Bias) were used to quantitatively evaluate the accuracy of time series of the TRMM and GPM satellite precipitation products. The formulas of the indices are as follows:

$$\text{CC} = \frac{\sum_{i=1}^{n}(M_i - \overline{M})(P_i - \overline{P})}{\sqrt{\sum_{i=1}^{n}(M_i - \overline{M})^2}\sqrt{\sum_{i=1}^{n}(P_i - \overline{P})^2}} \tag{1}$$

$$\text{RMSE} = \sqrt{\frac{\sum_{i=1}^{n}(P_i - M_i^2)}{n}} \tag{2}$$

$$\text{ME} = \frac{\sum_{i=1}^{n}(P_i - M_i)}{n} \tag{3}$$

$$\text{MAE} = \frac{\sum_{i=1}^{n}|P_i - M_i|}{n} \tag{4}$$

$$\text{Bias} = \frac{\sum_{i=1}^{n}P_i}{\sum_{i=1}^{n}M_i} - 1 \tag{5}$$

where $M_i$ is gauged rainfall at time $i$, mm; $\overline{M}$ is the average of $M_i$, mm; $P_i$ is satellite precipitation data at the same time, mm; $\overline{P}$ is the average of $P_i$, mm; and $n$ is the total number of samples.

Correlation Coefficient (CC) reflects the linear correlation between satellite rainfall and gauged rainfall [34]. The larger the value is, the better the correlation between the satellite rainfall data and measured rainfall data is. Mean Error (ME) reflects the average difference between the satellite rainfall and measured rainfall, while Mean Absolute Error (MAE) represents average amplitude of the error. Although the Root Mean Squared Error (RMSE) also measures the size of the average error, it gives more weight to the larger error. Relative Bias (Bias) reflects the systematic bias of satellite rainfall data.

Three more indices, namely Probability of Detection (POD), False Alarm Ratio (FAR), and Critical Success Index (CSI), were selected to assess accuracy of spatial patterns of rainfall. They are calculated as follows:

$$\text{POD} = \frac{H}{H + M} \tag{6}$$

$$\text{FAR} = \frac{F}{H + F} \tag{7}$$

$$\text{CSI} = \frac{H}{H + M + F} \tag{8}$$

where $H$ is the number of rainfall events observed and detected; $M$ is the number of rainfall events observed but not detected; and $F$ is the number of rainfall events detected but not observed. If the False Alarm Ratio (FAR) is smaller, the Probability of Detection (POD) and Critical Success Index (CSI) are higher and the satellite rainfall data have a better ability to estimate rainfall process.

### 3.2. Merging Method of Satellite Rainfall and Gauged Rainfall

The above evaluation indices are calculated for both the TRMM and GPM rainfall data at 3-hour time scales in the selected flood periods. Results as shown in the following Tables 1 and 2, indicating that the accuracy of both the TRMM and GPM rainfalls is unsatisfactory at hourly time scale; thus, the direct use of these satellite data is difficult to meet the requirements of flood forecasting due to amplification effect of error in rainfall-runoff calculation. A method of merging satellite rainfall and gauged rainfall is developed to improve precision of rainfall monitoring and the potential hydrological applications in medium and small catchments. Firstly, the spatial resolution of GPM satellite rainfall and gauged rainfall are interpolated to $0.25° \times 0.25°$, which is same with the spatial resolution of TRMM satellite rainfall. Compared to the Quantile Mapping (QM), this method is simpler and more intuitive [35]. The merging steps are as follows:

(1) Calculate the correction index: Taking the ratio of the measured rainfall to the satellite rainfall as the index $a_{i,j}$, the index $a_{i,j}$ is calculated for each grid at each time.

$$a_{i,j} = \frac{M_{i,j}}{P_{i,j}} \tag{9}$$

where $P_{i,j}$ is satellite rainfall of grid $j$ at time $i$, mm and $M_{i,j}$ is gauged rainfall of grid $j$ at time $i$, mm. If $P_{i,j} = 0$, then the gauged rainfall was directly taken as the final merging rainfall.

(2) Initial merging correction: An initial step size is selected for the index $a_{i,j}$, so the range of $a_{i,j}$ is divided into many intervals. Then, the hourly-scale satellite rainfall in each interval is multiplied by the median value of the corresponding interval to complete the initial merging correction.

(3) Determine appropriate step size of $a_{i,j}$: Reduce the step size of $a_{i,j}$ regularly, repeat steps (2) and (3), and obtain the correlation coefficient between measured rainfall and corrected satellite rainfall under different step sizes. When the correlation coefficient is larger than 0.9, the step size is considered as the appropriate one and the loop is stopped.

(4) The merging correction results: With the step size determined in the previous step, the corrected hourly-scale rainfall data is produced by merging the satellite and gauged rainfalls.

### 3.3. Evaluation of Merged Rainfalls

#### 3.3.1. Rainfalls Evaluation in the Xin'an River Catchment

The accuracy of TRMM and GPM satellite rainfall data are evaluated for the Xin'an River catchment before and after merging gauged rainfall, respectively. The results are shown in Table 1. The Correlation Coefficients (CC) between the measured rainfall and TRMM/GPM satellite rainfall before correction are all over 0.7, which indicates that these satellite rainfall products can reflect the rainfall processes in the

Xin'an River Catchment From the Mean Errors (ME); it can be seen that both TRMM 3B42 RT and V7 satellite rainfall underestimate the rainfall amounts in the flood events while the GPM IMERG satellite products are obviously overestimated. The Mean Absolute Error (MAE) and Root Mean Square Error (RMSE) also indicate that the satellite rainfall products still have obvious errors in monitoring the heavy rainfall events.

The evaluation indices PODs and CSIs are as low as 39%–70% for both TRMM and GPM satellite rainfall products, while False Alarm Ratios (FARs) are larger than 30% for the GPM rainfalls, and larger than 50% for the TRMM rainfalls. Generally, the GPM rainfalls show a better ability to capture the rainfall events; both the PODs and CSIs are slightly higher than the values of TRMM rainfalls. With the gauged rainfalls merged by the proposed method, the PODs and CSIs of the TRMM merged rainfalls are larger than 70%, while these indices become larger than 80% for the GPM merged rainfalls. The FARs are also significantly reduced to 16%–84%.

The plot diagrams of the satellite rainfalls and gauge rainfall are shown in Figure 2. The bias of the satellite rainfalls is effectively reduced at an hourly scale, especially for the heavy rainfall events, for which the plots scatter nearby 1:1 line. The merging method significantly improves the overestimation of GPM rainfalls and the underestimation of TRMM rainfalls in the Xin'an River catchment. As for small rainfall events, the merging method has a better effect for the GPM rainfall than the TRMM rainfalls.

**Table 1.** Accuracy of satellite rainfall before and after merging gauged rainfall in the Xin'an River catchment.

| Indices | TRMM 3B42RT | | TRMM 3B42 V7 | | GPM IMERG Early | | GPM IMERG Late | |
|---|---|---|---|---|---|---|---|---|
| | **Before** | **After** | **Before** | **After** | **Before** | **After** | **Before** | **After** |
| CC | 0.71 | 0.96 | 0.70 | 0.96 | 0.77 | 0.94 | 0.79 | 0.96 |
| RMSE | 2.24 | 0.91 | 2.70 | 0.98 | 3.38 | 0.48 | 3.64 | 0.37 |
| ME | −0.54 | −0.32 | −0.79 | −0.28 | 1.74 | 0.17 | 1.91 | 0.13 |
| MAE | 1.00 | 0.38 | 1.23 | 0.36 | 1.84 | 0.24 | 1.98 | 0.18 |
| Bias | −38% | −22% | −45% | −16% | 326% | 32% | 357% | 24% |
| POD | 41% | 72% | 46% | 78% | 64% | 80% | 70% | 84% |
| FAR | 59% | 28% | 54% | 22% | 36% | 20% | 30% | 16% |
| CSI | 39% | 72% | 44% | 78% | 56% | 80% | 62% | 84% |

Notation: CC—Correlation Coefficients; RMSE—Root Mean Square Error; ME—Mean Errors; MAE—Mean Absolute Error; POD—Probability of Detection; FAR—False Alarm Ratio; CSI—Critical Success Index.

**Table 2.** Accuracy of satellite rainfall before and after merging gauged rainfall in the Wuding River catchment.

| Indices | TRMM 3B42RT | | TRMM 3B42 V7 | | GPM IMERG Early | | GPM IMERG Late | |
|---|---|---|---|---|---|---|---|---|
| | **Before** | **After** | **Before** | **After** | **Before** | **After** | **Before** | **After** |
| CC | 0.43 | 0.93 | 0.42 | 0.93 | 0.48 | 0.92 | 0.42 | 0.91 |
| RMSE | 0.77 | 0.31 | 0.78 | 0.31 | 0.35 | 0.12 | 0.36 | 0.12 |
| ME | −0.15 | −0.02 | −0.16 | −0.02 | 0.50 | −0.01 | 0.42 | 0.00 |
| MAE | 0.23 | 0.08 | 0.23 | 0.09 | 0.09 | 0.04 | 0.11 | 0.04 |
| Bias | −61% | −7% | −63% | −8% | 2% | −6% | 10% | 0% |
| POD | 58% | 79% | 59% | 75% | 71% | 88% | 78% | 92% |
| FAR | 42% | 21% | 41% | 25% | 29% | 12% | 22% | 8% |
| CSI | 54% | 79% | 55% | 75% | 69% | 88% | 76% | 92% |

**Figure 2.** Comparison of satellite rainfalls before and after merging with gauged data in the Xin'an River Catchment.

### 3.3.2. Rainfalls Evaluation in the Wuding River Catchment

At hourly time scale, we also evaluate the accuracies of TRMM 3B42 and GPM IMERG satellite rainfalls before and after merging gauged data in the Wuding River catchment. The statistic indices are shown in Table 2. Compared to the results of the Xin'an River catchment in Table 1, the performances of both the TRMM and GPM satellite rainfall products are worse in the Wuding River catchment than that in the Xin'an River catchment. The Correlation Coefficients (CCs) are only larger than 0.4, which indicates that the correlation between satellite rainfalls and measured rainfalls is poor. The Mean Error (ME), Mean Absolute Error (MAE), and Root Mean Square Error (RMSE) are smaller than those in the Xin'an River catchment. The overestimation of GPM rainfall products is not obvious compared with that in Xin'an River catchment. The TRMM rainfalls still have a large Relative Bias (Bias), which shows that their monitoring accuracy is not high in the Wuding River catchment, while the performance of GPM is better.

In terms of spatial monitoring capacity, both PODs and CSIs of the GPM satellite rainfalls are higher than those of the TRMM satellite rainfalls and the FARs of the GPM satellite rainfalls are lower

than the values of the TRMM satellite rainfalls. It can be seen that GPM satellite rainfall products have better performance for small rainfall monitoring. For both the TRMM and GPM rainfalls merged with gauged rainfalls, all the indices are obviously improved, which indicates that the merging method is effective in integrating the advantages of satellite and gauged rainfalls at an hourly scale.

The plot diagrams in Figure 3 also indicate underestimations of both the TRMM and GPM rainfalls in the Wuding River catchment, although the latter also overestimate rainfall at some periods. Generally, maximum rainfall in the Wuding River catchment is relatively smaller than the value in the Xin'an River catchment during the study period. After merging with gauged rainfall, bias of the initial satellite rainfalls is significantly reduced. For the small rainfall events, the TRMM products seem to be corrected more effectively in the Wuding River catchment than the Xin'an River catchment.

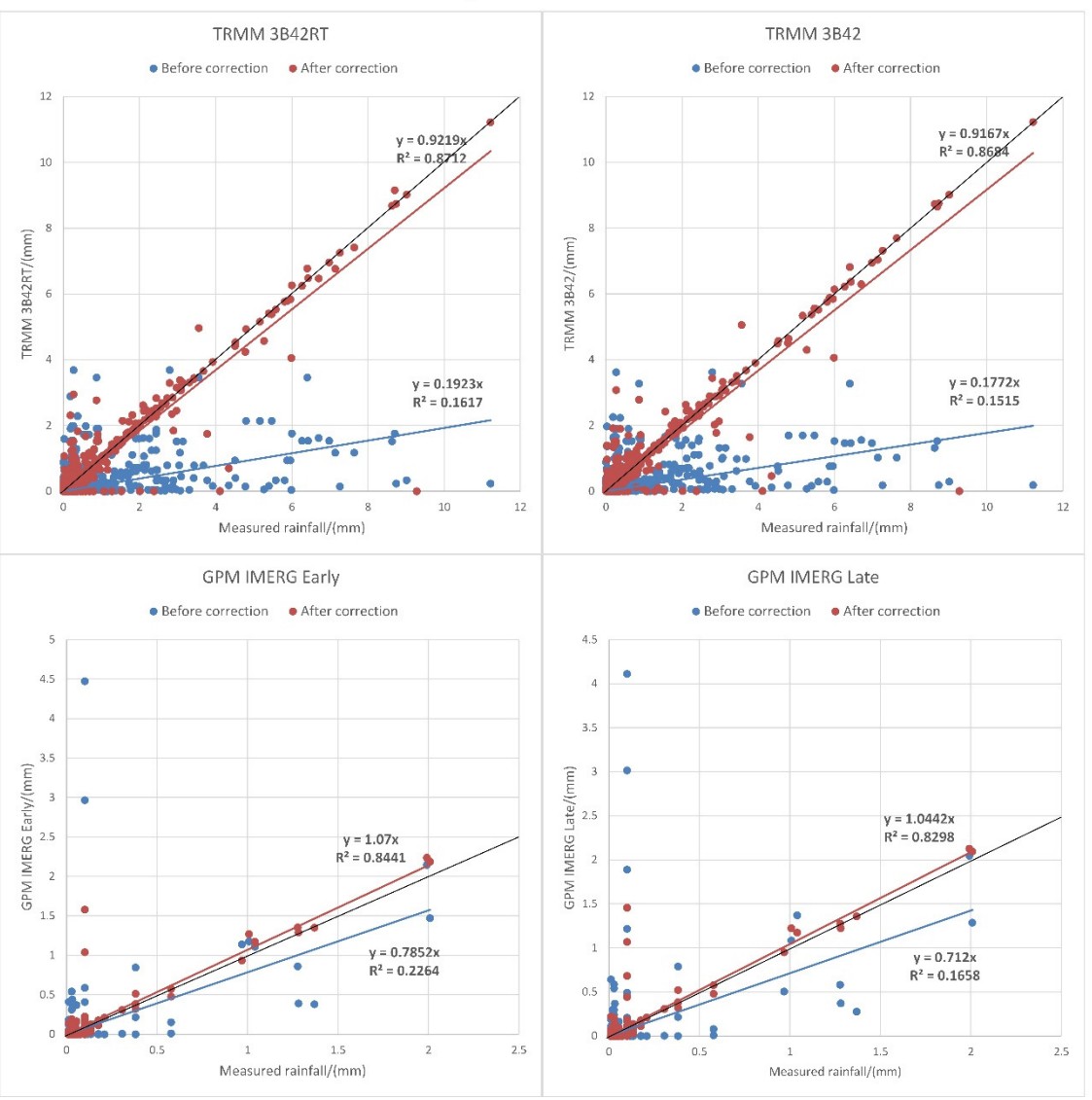

**Figure 3.** Comparison of the satellite rainfalls before and after merging with gauged data in the Wuding River catchment.

## 4. Improvement of Flood Simulation by the Merged Rainfall

### 4.1. HEC-HMS Model

In this study, the HEC-HMS (The Hydrologic Engineering Center's-Hydrologic Modeling System) model was used to simulate the floods processes both in the Xin'an River and Wuding River catchments. The rainfall-runoff model was developed by the Water Resources Research Center of the United States Army Engineering Corps in the 1990s [36]. For setting up the model, the SRTM 3 arc seconds DEM (Digital Elevation Model) is used by HEC-geoHMS to divide sub-catchments and relative parameters [37] for both the Xin'an River catchment and the Wuding River catchment. Hydrologic calculations of the model can be divided into four modules: runoff calculation, overland routing calculation, base flow calculation, and channel flow routing calculation [38]. For each module, the model provides several optional methods. The best suitable method can be selected, and the parameters will be validated according to the observed streamflow. In this study, SCS Curve Number Method, Synder Unit Hydrograph Method, Recession Method, and Muskingum Method were selected as the methods for each module of the model, which are used to calculate runoff and flow routing. The sensitive parameters include Lag Time, Muskingum K, and x. According to observed streamflow series, the model parameters were automatically optimized by the HEC-HMS model forced with gauged rainfall data.

### 4.2. Floods Simulated in the Xin'an River Catchment

Forced by gauged hourly rainfalls, the average Nash–Sutcliffe efficiency coefficients of simulated streamflow are 0.80 and 0.78 at the Linxi and Tunxi stations in the Xin'an River catchment, respectively, and the average correlation coefficients are 0.93 and 0.92, respectively. It can be seen that the accuracy of simulated result is good, which indicates that validated parameters of the model are available for the two catchments. The TRMM and GPM satellite rainfalls before and after merging gauged rainfalls were used to simulate the streamflow for the same rainfall-runoff events with the validated parameters. The accuracy of the simulated streamflow was evaluated as shown in Table 3.

**Table 3.** Accuracy of the simulated streamflow forced by satellite rainfalls before and after merging gauged rainfalls in the Xin'an River catchment.

| Stations | Rainfalls | TRMM 3B42RT | | TRMM 3B42V7 | | GPM IMERG Early | | GPM IMERG Late | | Gauged Rainfall | |
|---|---|---|---|---|---|---|---|---|---|---|---|
| | Indices | NS | CC | NS | CC | NS | CC | NS | CC | NS | CC |
| Linxi | Before | −0.01 | 0.61 | −0.09 | 0.50 | −39.83 | 0.63 | −48.78 | 0.64 | 0.80 | 0.93 |
| | After | 0.44 | 0.90 | 0.54 | 0.92 | 0.57 | 0.85 | 0.47 | 0.87 | | |
| Tunxi | Before | −0.25 | 0.80 | −0.36 | 0.81 | −9.32 | 0.88 | −9.59 | 0.89 | 0.78 | 0.92 |
| | After | 0.80 | 0.92 | 0.79 | 0.92 | 0.80 | 0.91 | 0.78 | 0.91 | | |

The averaged statistical indices of flood events simulated with different rainfall data indicate that the accuracy of streamflow simulated by the HEC-HMS model forced by the original satellite rainfalls is poor, and the simulations have negative Nash–Sutcliffe efficiency coefficients. In particular, because the original GPM satellite products significantly overestimate rainfall volumes in the Xin'an River catchment, the Nash–Sutcliffe efficiency coefficients of streamflow simulated with original GPM rainfall are even less than −40. Meanwhile, the correlation coefficients are all larger than 0.5, which indicate that there is a certain linear correlation between the simulated and observed streamflow. At the Tunxi station, the simulated streamflow is roughly the same as that at the Linxi station but the correlation coefficient is slightly larger than that of the Linxi station.

The accuracy of the streamflow simulated with the satellite rainfall after merging gauged data is improved obviously. At the Linxi station, the Nash–Sutcliffe efficiency coefficients are all about 0.5 and the correlation coefficients are about 0.9. GPM satellite rainfalls have a significant correction effect. At the Tunxi station, the Nash–Sutcliffe efficiency coefficients of the simulated streamflow are

about 0.8 and the correlation coefficients are larger than 0.9, which indicate that the merged GPM satellite rainfalls have a good performance to be used in flood simulations. Correlation coefficients of the streamflow simulated with the TRMM 3B42RT and GPM IMERG Early rainfalls after merging gauged rainfall are even larger than the results simulated only with the gauged rainfall at the Tunxi station. This proves that the merging method has a potential ability to integrate advantages of both satellite rainfall and gauged rainfall.

Hydrographs of four typical flood events at Linxi station are shown in Figure 4. Although the streamflow processes are similar in time series, the TRMM satellite rainfalls before merging gauged rainfall obviously underestimate most of the flood peaks, especially for the single-peak flood events, such as Flood 20120626 and Flood 20150607. For Flood 20110608 with continuous flood peaks, the simulated streamflow with TRMM satellite rainfalls agrees well with the observed streamflow. However, the simulated streamflow with the GPM satellite rainfall before merging gauged rainfall is too large for hydrological application in the catchment. The reason is that the overestimation of GPM satellite rainfall has a great impact on flood forecasting accuracy, although the peak times show a good agreement with the measured values. Both the original TRMM 3B42RT and 3B42V7 rainfalls underestimate rainfall volumes in the four flood periods; therefore, the streamflow is also significantly underestimated. The hourly hydrographs simulated with the TRMM 3B42V7, which has been assimilated with daily rainfalls of sparse meteorological stations, show a slightly better performance at the Linxi station than the result simulated by the TRMM 3B42RT, while GPM IMERG Early and GPM IMERG Late have no obvious difference due to no gauged rainfall assimilated into them. After merging gauged rainfall, the streamflow simulated with TRMM and GPM satellite rainfalls fits well with the observed floods and performs well in aspects of flood peak, flood volume, and flood process.

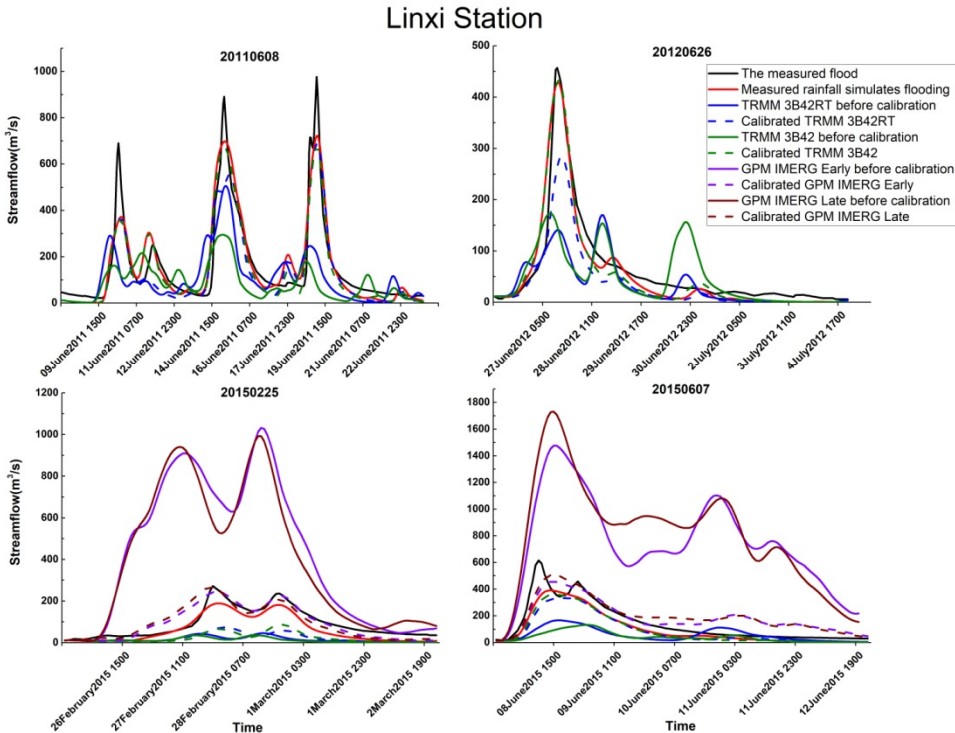

**Figure 4.** Streamflow simulated with different rainfall products at the Linxi station.

Similarly, four typical flood events are evaluated at the Tunxi station, as shown in Figure 5. Original TRMM 3B42 rainfalls also underestimate the heavy rainfall processes, while original GPM IMERG rainfalls underestimate rainfall of Flood 20150510 and overestimate rainfall of Flood 20150607. Therefore, the hydrographs simulated with these original satellite rainfalls show obvious bias compared

with the observations. When forced by the satellite rainfalls after merging gauged rainfall, the HEC-HMS model simulates the streamflow pretty well compared to the observed hydrograph at the Tunxi station. It can be concluded that the satellite rainfalls corrected by the merging method are suitable for flood forecasting in humid regions.

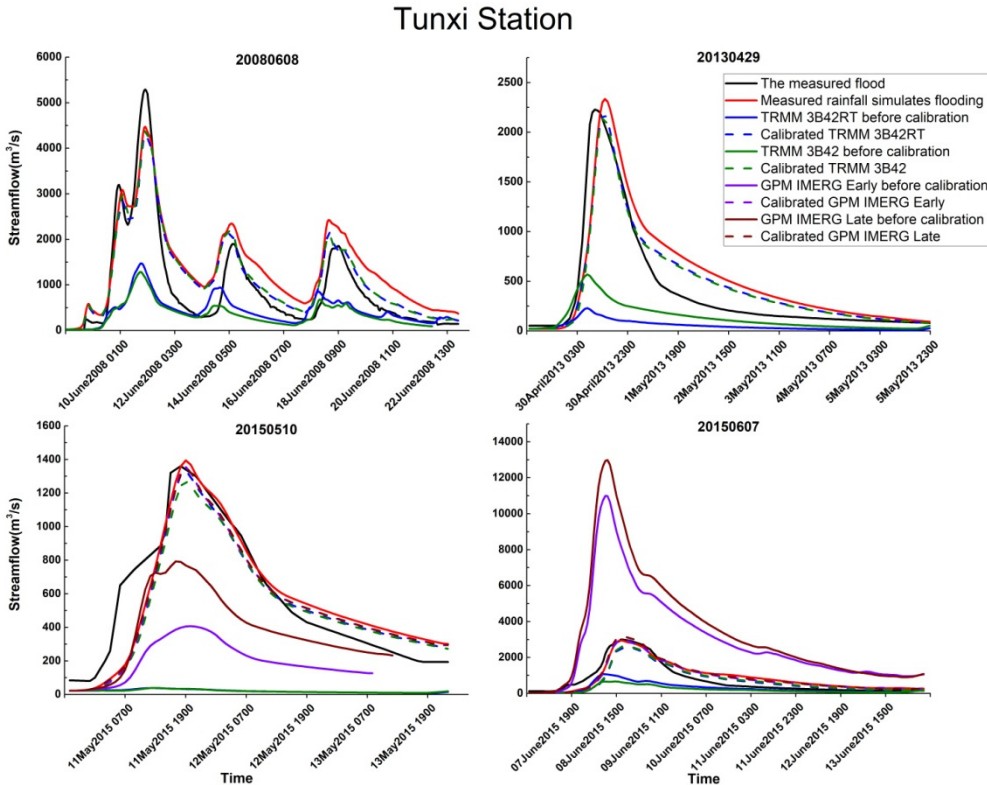

**Figure 5.** Streamflow simulated with different rainfall products at the Tunxi station.

### 4.3. Floods Simulated in the Wuding River Catchment

When forced with the observed hourly rainfalls, HEC-HMS model can also provide acceptable simulation results of floods at the semiarid Wuding River catchment after carrying out a calibration process of model parameters. The average Nash–Sutcliffe efficiency coefficients are 0.78 and 0.70 at Qingyangcha and Suide stations, respectively, and the correlation coefficients are 0.89 and 0.87, respectively. With the calibrated parameters, the streamflow was simulated by the model forced by the TRMM 3B42 and GPM IMERG satellite rainfalls before and after merging gauged rainfall, respectively. The results are shown in Table 4.

According to the averaged statistical indices in Table 4, we can see that the hydrology model only outputs poor simulation results when forced by the original TRMM and GPM satellite rainfalls in the Wuding River catchment. The Nash–Sutcliffe efficiency coefficients range between −11.58 and 0.11, and the correlation coefficients ranging between 0.23 and 0.78. GPM IMERG rainfalls show a better performance in the Wuding River catchment than the results in the Xin'an River catchment in the Table 3 because the overestimation of GPM rainfalls has been well addressed in the Wuding River catchment. Meanwhile, GPM rainfalls seem to have larger correlation coefficients than the TRMM rainfalls, which indicates that GPM rainfall can better monitor rainfall processes.

**Table 4.** Accuracy of the simulated streamflow forced by satellite rainfalls before and after merging gauged rainfalls in the Wuding River catchment.

| Station | Rainfalls | TRMM 3B42RT | | TRMM 3B42 V7 | | GPM IMERG Early | | GPM IMERG Late | | Gauged Rainfall | |
|---|---|---|---|---|---|---|---|---|---|---|---|
| | Indices | NS | CC | NS | CC | NS | CC | NS | CC | NS | CC |
| Qingyangcha | Before | −0.04 | 0.37 | −0.02 | 0.43 | 0.11 | 0.67 | 0.02 | 0.23 | 0.78 | 0.89 |
| | After | 0.63 | 0.88 | 0.59 | 0.88 | 0.52 | 0.86 | 0.50 | 0.87 | | |
| Suide | Before | −0.31 | 0.38 | −0.46 | 0.37 | −5.92 | 0.85 | −11.58 | 0.78 | 0.70 | 0.87 |
| | After | 0.49 | 0.85 | 0.32 | 0.84 | 0.60 | 0.90 | 0.45 | 0.89 | | |

Accuracy of the streamflow simulated with satellite rainfalls after merging gauged rainfalls is significantly improved at both Qingyangcha and Suide stations. The Nash–Sutcliffe efficiency coefficients are about 0.5, and all the correlation coefficients are larger than 0.8. It can be concluded that the merging method also has feasibility in the Wuding River catchment and can greatly improve the accuracy of flood forecasting when satellite rainfalls used.

Hydrographs of four typical flood events are shown in Figure 6. Minor flood event, i.e., Flood 20110729, cannot be simulated by the HEC-HMS model with the satellite rainfall. In most cases, the TRMM satellite products underestimate the rainfall, such as in Floods 20070725 and 20070804. While the GPM satellite products show an unstable performance in the Wuding River catchment, they also underestimate rainfall in the period of Flood 20140720. Compared to the results in the Xin'an River catchment, the satellite products perform a bit worse in the Wuding River catchment and cannot have same good monitoring capability to small rainfalls anymore.

The merging method not only improved underestimation of TRMM satellite rainfall products in Flood 20070804 but also corrected the false and missing observation records in Floods 20070725 and 20110729. The streamflow simulated with the satellite rainfalls after merging gauged data fits well with the actual hydrographs at Qingyangcha station.

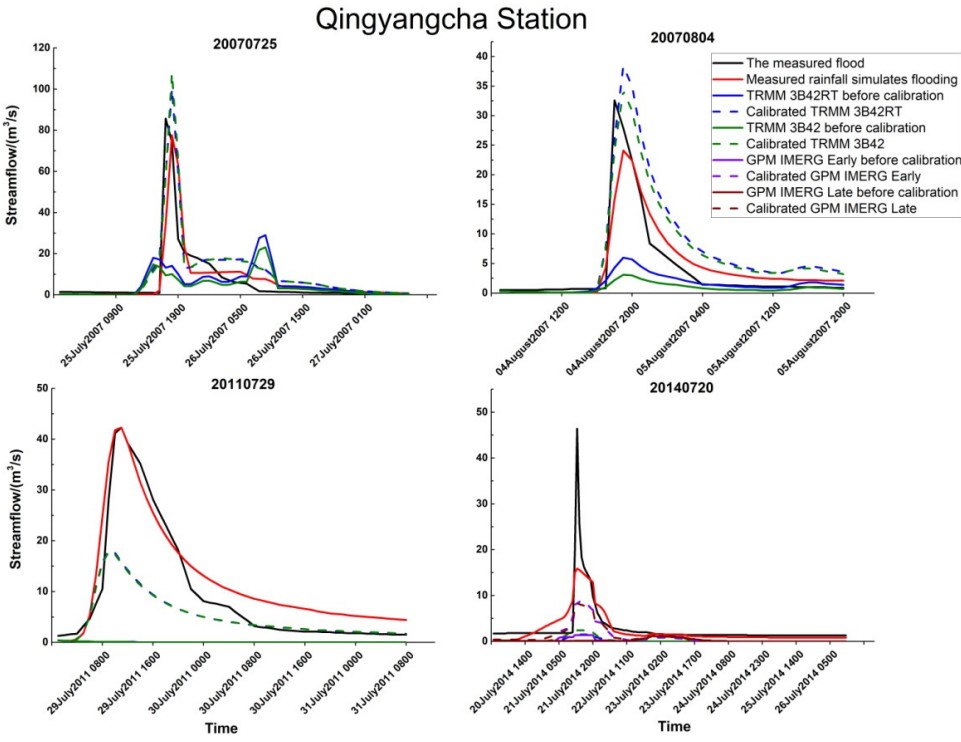

**Figure 6.** Streamflow simulated with different rainfall products at Qingyangcha station.

Simulated hydrographs of four typical floods at Suide station are shown in Figure 7. Similar to the results in the Xin'an River catchment, TRMM satellite rainfalls underestimate rainfall volume.

The GPM satellite rainfalls overestimate the rainfalls in most cases, but there are also some cases underestimated by the GPM, e.g., Flood 20140721. However, the TRMM satellite data even fail to reproduce this flood, which shows that the satellite is not sensitive enough to monitor small rainfalls in the Wuding River catchment. Also, streamflow simulated with the original TRMM rainfall has some problems in capturing the peaking time. After merging the gauged rainfall, simulated streamflow has significantly improved modeling capacity of the HEC-HMS at aspects of both flood volume and peak time.

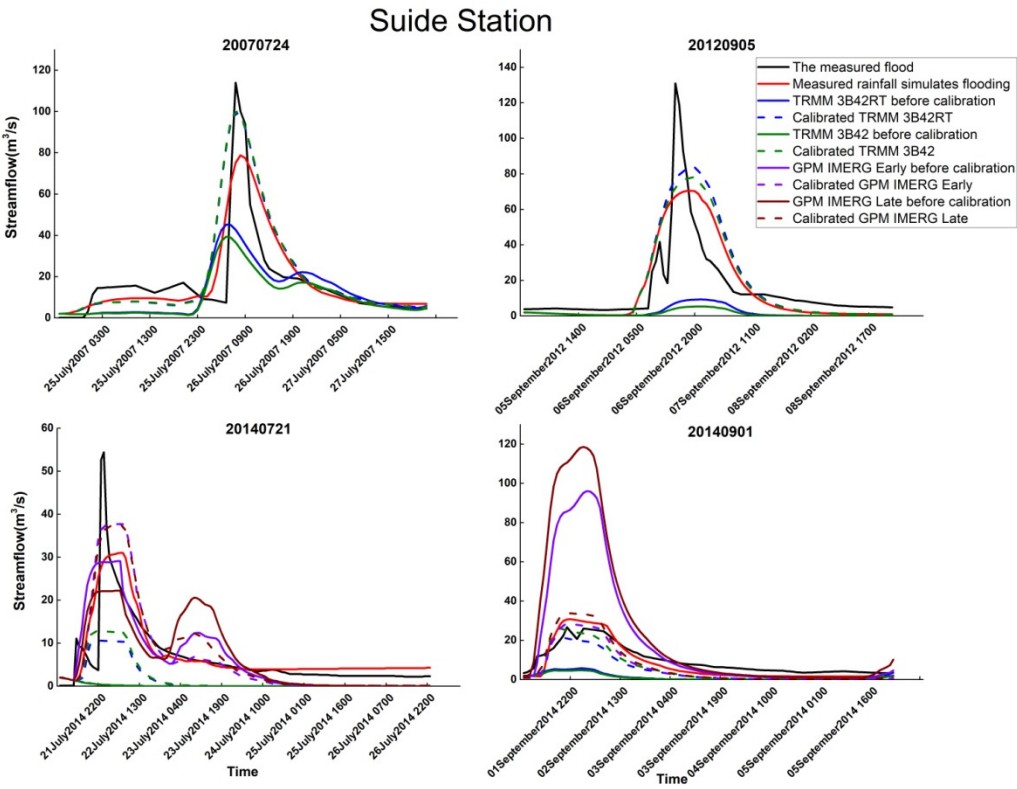

**Figure 7.** Streamflow simulated with different rainfall products at the Suide station.

## 5. Discussion and Conclusion

In this study, TRMM 3B42 and GPM IMERG satellite rainfalls were evaluated at hourly scales for periods of the flood events in the Xin'an River and Wuding River catchments. At such small temporal and spatial scales, the original satellite rainfalls usually overestimate or underestimate actual rainfall processes in the study regions. When these data are used for flood simulations in small–medium catchments, they need to be corrected by gauged rainfalls. Therefore, an error correction method of merging satellite rainfall and gauged rainfall was developed to improve precision of rainfall monitoring. Due to limitations of both satellite monitoring data and hydrology data, we can only choose more than ten flood events for each hydrology station of the two catchments in this study. HEC-HMS is chosen to simulate the floods because the hydrologic model has been verified to be applicable in both humid and arid regions. With the merged rainfalls, streamflow simulation by the model was significantly improved.

Despite the correlation coefficients being all larger than 0.7 with the gauged rainfall, original TRMM and GPM rainfall products have only a limited ability to capture the real precipitation in the humid Xin'an River catchment given that the TRMM rainfall products underestimate rainfall amounts, and the averaged bias is about −40%. On the contrary, the GPM rainfall products have an obvious overestimation of local rainfall. Generally, the GPM rainfalls have better performance according to the evaluation indices.

In the semiarid Wuding River catchment, precision of both TRMM and GPM rainfalls is lower than that in the Xin'an River catchment and the correlation coefficients of satellite rainfalls with the gauged rainfall are only about 0.4. The underestimation of the TRMM rainfall is still obvious and its bias is less than −60%, while the GPM rainfall shows a better performance with a bias of less than 10%.

The merging method of satellite rainfall and gauged rainfall provides a tool to integrate the two rainfall data sources. It significantly solves the overestimation and underestimation problems of the satellite rainfalls. In addition, missing measurement and false alarm of satellite rainfall products are also improved, which greatly improves accuracy of the satellite rainfall. In both the Xin'an River and Wuding River catchments, correlation coefficients between merged rainfall and gauged rainfall are all larger than 0.9 and the POD is about 80%, which indicates that the method has important application value for flood forecasting in ungauged areas.

In fact, accuracy of streamflow simulated with the original satellite rainfalls is poor in the studied catchments. The results cannot meet requirements of flood forecasting at hourly time scale. In both the Xin'an River and Wuding River catchments, the Nash–Sutcliffe efficiency coefficients of simulated streamflow by using the original TRMM and GPM satellite rainfalls are negative mainly due to the large errors of flood volume and peaks. The satellite products cannot monitor rainfall at some time during a flood event in the Wuding River catchment.

On the other hand, in the Xin'an River catchment, the Nash–Sutcliffe efficiency coefficients of streamflow simulated with the merged TRMM and GPM rainfalls are about 0.7 and 0.8, respectively, and all the correlation coefficients are about 0.9. In the Wuding River catchment, the Nash–Sutcliffe efficiency coefficients are about 0.5 and the correlation coefficients are about 0.85. Generally, the TRMM and GPM satellite products perform better in the Xin'an River catchment than the Wuding River catchment.

At present, satellite rainfall has been used in streamflow simulation in many basins at daily and even monthly time steps. The satellite rainfall products also show the potential to be used in flood forecasting in medium and small catchments at hourly time steps. It is necessary to further improve accuracy of satellite rainfall monitoring and to develop merging methods with more extensive applicability in the future. Therefore, satellite rainfall can be better applied in hydrology and water resources research.

**Author Contributions:** Conceptualization, C.Y.; Data curation, X.M. and N.D.; Formal analysis, X.M.; Funding acquisition, C.Y.; Methodology, X.M., C.Y. and N.D.; Project administration, C.Y.; Validation, X.M. and N.D.; Writing—original draft, X.M.; Writing—review & editing, C.Y. and N.D. All authors have read and agreed to the published version of the manuscript.

**Funding:** This research was funded by the National Key R&D Program of China (grant number 2016YFC0402706); Fundamental Research Funds for the Central Universities (grant number 2018B55114); Special Scientific Research Fund of the Meteorological Public Welfare Profession of China (grant number GYHY201406021); National Natural Science Foundation of China (grant number 41471016, 51539003); and Special Fund of State Key Laboratory of Hydrology-Water Resources and Hydraulic Engineering (grant number 20185044012).

**Conflicts of Interest:** The authors declare no conflict of interest.

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
