# Peer review of "Merging Satellite and Gauge Rainfalls for Flood Forecasting of Two Catchments Under Different Climate Conditions"

_water, doi:10.3390/w12030802_

Round 1

Reviewer 1 Report

This study evaluates TRMM and IMERG satellite datasets in terms of their suitability as inputs for flood forecasting. For this purpose, flood simulations are performed over two catchments in China. In addition, the authors apply a gauge merging method to the satellite datasets and find that the gauge-merged data show better performance for the flood simulations. The study topic is of interest to the readership of Water. However, before being publishable in the journal, the following comments should be addressed.

General comments

It is not clear why the authors use satellite-only version IMERG, i.e. Early and Late runs. There are already gauge-corrected version (i.e., Final run) available. This should be also the case for TRMM. What is the main advantage of using the gauge merging method proposed in the study? In the study, the gauge-merged satellite data lead to significant improvement in streamflow simulations. In other words, we (still) can not guarantee the quality of satellite-only data when there are no gauge data and the merging method can not be applied. In this context, is it still valid that i) satellite data is an important mean to solve the problem of “ungauged” basins (see Abstract), ii) indicates a good application prospect of satellite rainfall in hydrological forecasting (see Abstract), or iii) the method has important application value for flood forecasting in “ungauged” areas (see Conclusions) ? The main results (e.g., the accuracy of satellite data) are often compared with respect to humid vs arid conditions. I understand that the two catchments are located in different climatic regions. But, given only two catchments are considered in this study, I think it is hard to evaluate the performance of the satellite data in terms of humid/arid conditions; i.e., the performance differences between the two catchments can be due to many other factors, not due to humid vs arid conditions only. 

Specific comments

Please introduce the full names of abbreviations (e.g., TRMM, GPM, CMORPH, THREW, etc.) at the beginning. Sect. 2.2; provide more details on the satellite data. For instance, what is the difference between TRMM 3B42 V7 and 3B42RT, what is the difference between IMERG early and late?  Sect. 2.2; there are several errors.  TRMM means “Tropical Rainfall Measuring Mission”, not “Tropical rainfall Measurement Program” GPM means “Global Precipitation Measurement “, not “Global precipitation Measurement Program” Some GPM IMERG versions are available from 2000, not only available after 2014. TRMM is the first satellite missions “dedicated to measuring tropical and subtropical rainfall”  Please provide proper references for TRMM and GPM missions. 

Minor comments

Abstract; the problem of prediction => prediction of what?  Abstract; in most cases, the “efficiency coefficients” of the simulated streamflow … => does the efficiency coefficients mean “Nash–Sutcliffe efficiency coefficients”? Please make it clear. P2; rainfall data is not available due to the change of topography and climate => it is not clear why the data are not available due to topography or climate, please make it clear. P2; the spatial resolution and precision of the GPM satellite rainfall is higher than what? P2; GPM satellite has more advantage => please explain “advantage” P2; the results showed that the adjusted data => do you mean “gauge-adjusted data”? P3; The TRMM and GPM satellite … greatly impacted by local climate and terrain conditions => please provide references.  P3; The precision analysis of hourly satellite rainfall is less studied => there are still some studies, see, e.g., 10.1175/JHM-D-16-0174.1 , 10.1002/qj.3218 P3; can you define “large” vs “medium and small” basins? How small are your catchments compared to the previous studies listed in the introduction?  Fig1. It would be nicer if longitude/latitude information is shown in the map.  P5; in eq.9, does the satellite pixel (P) have at least one gauge (M) always? What if there is no gauge or multiple gauge in the same pixel - how did you calculate the index a? P6; and repeat steps (3) and (4) => there is no step (4)  P9; SCS Curve Number method, Synder Unit Hydrograph Method, Recessions Methods and Muskingum Methods => what are these methods? Are they methods selected for each module of the model? Please make it clear. 

Author Response

Dear reviewer:

        We have carefully revised our manuscript based on your comments, and please see the attachment.Thank you.

Reviewer 2 Report

There are a number of cases where an article should be used with a noun and are not. There are other cases where an article is used in the text but would not be used in English. Also, there are some cases on Non-English or strange English word usages. 

The meaning of the text is clear. The less than usual usages are not terrible but the article would be improved in a textual manner if corrections were made.

Author Response

Dear reviewer:

        We have studied the valuable comments from you carefully, and tried our best to revise the manuscript.Please see the attachment,thank you.

Round 2

Reviewer 1 Report

In this revision the authors made adequate improvements of the paper. Just one minor thing - please change CMORPH(CPC MORPHing technique) to CMORPH (Climate Prediction Center morphing method) in Page 2. See https://doi.org/10.1175/1525-7541(2004)005<0487:CAMTPG>2.0.CO;2